# Eight-Element Dual-Band Multiple-Input Multiple-Output Mobile Phone Antenna for 5G and Wireless Local Area Network Applications

**DOI:** 10.3390/mi14122200

**Published:** 2023-11-30

**Authors:** Tao He, Jianlin Huang, Jiaping Lu, Xiaojing Shi, Gui Liu

**Affiliations:** College of Electrical and Electronic Engineering, Wenzhou University, Wenzhou 325035, China; 21451841012@stu.wzu.edu.cn (T.H.); 194511981414@stu.wzu.edu.cn (J.H.); 22451841024@stu.wzu.edu.cn (J.L.); shi@wzu.edu.cn (X.S.)

**Keywords:** dual-band, fifth generation (5G), WLAN, MIMO antenna, mobile phone antenna, sub-6 GHz

## Abstract

This paper proposes an eight-element dual-band multiple-input multiple-output (MIMO) antenna that operates in the fifth generation (5G), n78 (3400–3600 MHz), and WLAN (5275–5850 MHz) bands to accommodate the usage scenarios of 5G mobile phones. The eight antenna elements are printed on two long frames, which significantly reduce the usage of the internal space of the mobile phone. Each antenna element is printed on both surfaces of one frame, which consists of a radiator on the internal surface and a defected ground plane on the outer surface. The radiator is a rectangular ring fed by a 50 Ω microstrip line which is printed on the top surface of the system board. A parasitic unit is printed on the outer surface of each frame, which is composed of an inverted H-shaped and four L-shaped patches. Each parasitic unit is connected to the internal surface of the frames through a via, and then it is connected to a 1.5 mm wide microstrip line on the top surface of the system board, which is connected to the ground plane on the bottom surface of the system board by a via. Four L-shaped slots, four rectangular slots, and four U-shaped slots are etched onto the system board, which provides good isolation between the antenna elements. Two merged rectangular rings are printed on the center of each frame, which improves the isolation further. The return loss is better than 6 dB, and the isolation between the units is better than 15 dB in the required working frequency bands. In addition, the use of a defected ground structure not only makes the antenna element obtain better isolation but also improves the overall working efficiency. The measurement results show that the proposed MIMO antenna structure can be an ideal solution for 5G and WLAN applications.

## 1. Introduction

In recent years, 5G communication technology has gradually become popular, which can provide high transmission frequencies and large bandwidths. Since the MIMO antenna can not only improve the spectral efficiency to a certain extent but also support each node to output more data, MIMO technology has been applied to the mobile communication system. Research on MIMO communication originated in the late 1990s [1]. Around 2006, MIMO technology was introduced into WiFi systems and 5G communication systems. The characteristics of multiple spatial dimensions of MIMO antennas can be used to improve signaling and spectral efficiency. Since 5G communication has to overcome the problems of delay, energy consumption, cost, and high data transmission rates, MIMO antennas have become an important research area. According to 3GPP, 5G new radio (N.R.) frequency bands are mainly divided into sub-6 GHz frequency bands and millimeter-wave frequency bands [2].

Currently, the antennas for 2G/3G/4G are typically integrated into the mainboard of mobile terminals [3,4,5,6]. It is a hot topic to design an antenna in a limited space and achieve the primary performance of a MIMO antenna. To ensure that 5G frequency bands can be covered without affecting the overall dimensions of the mobile phone, 5G antennas can be printed on both surfaces of the metal frames [7,8]. Research on the frequency bands of 5G MIMO antenna concentrates on the range of 3300–5000 MHz [9,10,11]. In 2015, 3400–3600 MHz was announced as a new 5G communication band at the World Radio Communication Conference, which is the first licensed 5G test frequency band in China. A MIMO antenna structure with orthogonal placement is presented in [9], which not only has excellent isolation but also covers frequency bands of 3100–3850 MHz and 4800–6000 MHz. In [10], a MIMO antenna composed of dual-antenna arrays was proposed, which covers the 5G New Radio bands of n77 (3300–4200 MHz), n78 (3400–3600 MHz), and n79 (4800–4900 MHz). A double-loop antenna unit structure is proposed in [12], which enables the antenna to cover 3400–3800 MHz and 4800–5000 MHz bands.

To achieve the design goals of 5G MIMO antennas, mobile terminals often need to install six to eight or even more antenna elements in the sub-6 GHz bands in a limited space. At present, there are antenna arrays with more than eight ports, which results in limited space for antennas of other frequency bands. These antennas are distributed on the dielectric substrate and occupy a significant amount of space. Such a distribution may not be suitable for widescreen mobile phones. In addition, due to the limitation of space, the coupling effect between the antenna elements is substantial. At present, many structures can make the antenna have good isolation. Polarization and pattern diversity are applied to optimize the isolation effect between two adjacent antenna elements [13,14,15,16]. Additionally, slot structure is a more common and effective decoupling method. References [17,18] achieve good isolation performance between two adjacent antenna elements by employing a slot configuration. In summary, how to design a wideband MIMO antenna with appropriate size and good isolation is a topic of concern at present.

This paper proposes an eight-element MIMO antenna for 5G mobile phones, which covers the frequency bands of 5G n78 (3400–3600 MHz) and WLAN band (5275–5850 MHz). The eight antenna elements are printed on two long frames of a mobile phone. An isolation structure is employed to solve the problem of poor isolation between the middle antenna elements. The return loss of the MIMO antenna is less than 6 dB, and the isolation is higher than 15 dB.

The remainder of this paper is organized as follows. The structure and size of the antenna are introduced. The evolution design of the antenna is presented. The S-parameter characteristics and the current distribution are simulated and analyzed. Finally, the measurement results of the prototype are discussed and analyzed. 

## 2. Antenna Structure

The detailed design and geometric structure of the proposed eight-element dual-band antenna array for 5G mobile phones are shown in Figure 1. The substrate of the two frames and the system board is FR4, with a thickness of 0.8 mm, a relative permittivity of 4.4, and a loss tangent of 0.02. The size of the system board and two long frames is 150 mm × 75 mm × 0.8 mm and 150 mm × 7 mm × 0.8 mm, respectively. The overall size of the presented antenna system meets the general dimension requirements of 5G mobile phones. As shown in Figure 1a, the distance between the bottom of the frames and the system board is 1.5 mm. The feed line is fed by a coaxial cable from the via on the ground plane. There are four antenna elements on each frame, and the distance between Ant. 1 and Ant. 2 is 32.5 mm, while the distance between Ant. 2 and Ant. 3 is only 30.5 mm. The distance between Ant. 3 and Ant. 4 is the same as the distance between Ant. 1 and Ant. 2. The four antenna elements (Ant. 5 to Ant. 8) on the other frame also present a symmetrical structure. Ant. 1–Ant. 4 and Ant. 5–Ant. 8 are symmetrical about the central axis. Each antenna element consists of a radiator and a parasitic unit, as shown in Figure 2. The parasitic unit is printed on the outer surface of the mobile phone frame, which is composed of an inverted H-shaped and four L-shaped patches. The radiator is printed on the inner surface of the mobile phone frame, which is a rectangular ring. In addition, the radiator is connected to the feed line with a width of 1.5 mm and a length of 8 mm. The parasitic unit is connected to the ground through the vias. To improve the isolation between the two antenna elements in the middle of each frame, a back-to-back placement and additional isolation units are used in this paper. In Figure 2c, the overall dimension of the isolation unit is 17.65 mm × 7 mm, which has two rectangular slots of 5.83 mm × 5 mm. One isolation unit is printed between Ant. 2 and Ant. 3 on one frame, while the other isolation unit is printed between Ant. 6 and Ant. 7 on the other frame. The isolation unit can attenuate the influence of the coupling between the antenna elements, thus achieving a better isolation effect.

## 3. Design Procedure and Analysis

### 3.1. Antenna Element Design

The S-parameters of the MIMO antenna and the electric field distribution on the radiation patch are analyzed, and the working mechanism of the MIMO antenna is explained. The simulated S-parameters of the antenna elements are shown in Figure 3. Since Ant. 1–Ant. 4 and Ant. 5–Ant. 8 are mirror images for the system ground plane, this section only discusses Ant. 1–Ant. 4. In Figure 3, the two resonant modes excited by the eight-element antenna can fully cover the lower-frequency and higher-frequency bands. The reflection coefficients of each antenna element are better than −6 dB. To demonstrate the excitation of these two resonant modes, Figure 4 shows the surface electric current density distributions at 3.5 GHz and 5.5 GHz, respectively. The current distributions at 3.5 GHz and 5.5 GHz indicate that the external patch has a significant effect on the generation of dual-frequency bands, and the existence of the internal rectangular frame can not only optimize the impedance matching of the antenna elements but also strengthen the current of the external patch. 

### 3.2. Design Procedure

Several structures and corresponding S-parameters in the antenna design process are shown in Figure 5 and Figure 6. The vias connected to the parasitic units and the feed lines are represented by red cylinders in Figure 5. As depicted in Figure 6, the −6 dB impedance frequency band of the Case 1 antenna is 3.1–4.33 GHz, which covers the n78 frequency band. However, the higher-frequency band does not meet the design requirements. In addition, the horizontal strips are too long, and the coupling effect between the antenna elements is relatively strong. The lower-frequency characteristics of the Case 2 antenna meet the design requirements, and the bandwidth of the higher-frequency ones is optimized. However, the resonant frequency of the higher-frequency band is far from 5.5 GHz. To optimize the resonant frequency of the higher-frequency band, Case 3 is designed. The resonant frequency of the higher-frequency band is decreased to 6.48 GHz. To further optimize the resonant frequency of the higher-frequency band, the Case 4 antenna is proposed. In this model, a rectangular strip is added to the right patch. The resonance point of the higher-frequency band reaches 5.49 GHz. In addition, the frequency band of the antenna covers n78 (3400–3600 MHz) and WLAN (5275–5850 MHz), which meets the design requirements.

The surface current distributions of antenna elements at different frequencies are shown in Figure 7. At 3.5 GHz, the current is mainly concentrated on the left side of the external patch, while the current distribution of the right patch is weaker. At 5.5 GHz, the current on the patch is mainly concentrated on the right side, while the current distribution on the left side is relatively weak. 

The S-parameters of the MIMO antenna without the isolation structure are illustrated in Figure 8. It can be observed that the return loss of the MIMO antenna without isolation structure can cover the n78 (3400–3600 MHz) and WLAN band (5275–5850 MHz). However, the isolation is lower than 15 dB in both frequency bands.

The S-parameters of the MIMO antenna with an isolation structure are illustrated in Figure 9. The isolation structure includes the reduction in the ground radiation patch and the addition of isolation cells. As seen in Figure 9, the return loss of the MIMO antenna is optimized and extended in frequency bands less than −6 dB after the addition of the isolation unit. The isolation of the antenna is larger than 15 dB in the required operating band. Therefore, the frequency band of the proposed MIMO antenna can cover n78 (3400–3600 MHz) and WLAN (5270–5850 MHz) frequency bands, and the simulated return loss and isolation meet the design requirements.

The dimensions of the slot on the top surface of the system board for two different shapes are depicted in Figure 10. Both the L-shaped slot and the rectangular slot are situated between Ant.5 and Ant.6. In Figure 1b, it can be observed that the inverted L-shaped slot and the rectangular slot share the same dimensions as their counterparts, the L-shaped slot and rectangular slot, respectively. These inverted L-shaped and rectangular slots are etched between Ant.1 and Ant.2, Ant.3 and Ant.4, as well as Ant.7 and Ant.8. Additionally, the inverted U-shaped slots mirror the dimensions of the U-shaped slot presented in Figure 10b. Figure 1a illustrates that both the U-shaped slots and the inverted U-shaped slots are etched between Ant.2 and Ant.3, as well as Ant.6 and Ant.7.

Figure 11 and Figure 12 show the current distribution without and with the isolation structure, respectively. As seen from the two current distribution cases, the isolation unit can significantly reduce the coupling current between Ant. 2 and Ant. 3.

### 3.3. Parameter Analysis

As shown in Figure 13, the envelope correlation coefficient (ECC) is below 0.024 in both frequency bands, which is lower than the threshold of 0.5 for the normal operation of a MIMO antenna. A low ECC value indicates that the antenna has a good MIMO correlation and better diversity performance. The ECC values are calculated by Equation (1).
(1)$$\rho_{c}(i,\;j)=\frac{{\left|{}_{4\pi }^{\iint }|\;[F_{i}(\theta ,\varphi )\cdot F_{j}^{*}(\theta ,\varphi )]\;d\Omega \right|}^{2}}{{}_{4\pi }^{\iint }{\left|F_{i}(\theta ,\varphi )\right|}^{2}d\Omega \;\cdot {}_{4\pi }^{\iint }\;{\left|F_{j}(\theta ,\varphi )\right|}^{2}d\Omega }$$

Figure 14 shows the diversity gain (DG) of the proposed antenna, which is higher than 9.99 dB in both operating bands. Therefore, the antenna has good anti-jamming ability and low signal-to-noise ratio fluctuation.

## 4. Results and Discussion

The results in Figure 15 illustrate the performance of the MIMO antenna in a practical application scenario. Figure 15a,b show the influence of the human hand on the antenna current distribution at 3.5 GHz and 5.5 GHz. Figure 15c,d show the influence of the human body on the antenna S-parameters, respectively. In the operating frequency bands, the antenna exhibits a return loss better than −6 dB, and the isolation of the antenna is better than 15 dB. The performance of the antenna meets the design requirements.

We preliminarily welded the antenna according to the design size. As shown in Figure 16, each radiation unit of the eight-port antenna is connected to the SMA connector at the bottom of the antenna via a microstrip line. In addition, Figure 17 shows the actual measurement operation scenario.

The S-parameters were measured using the vector network analyzer (VNA) instrument. The measured and simulated parameters are shown in Figure 18. The measured results roughly match the simulation results, and small model errors are observed due to manufacturing process issues. It can be seen from the figure that Ant. 1–Ant. 4 show good parameter characteristics in the n78 (3400–3600 MHz) and WLAN band (5275–5850 MHz). It is worth noting that the isolation between Ant. 1 and Ant. 4 in the actual measurement is better than the simulation characteristics, and isolation is better than 15 dB in all frequency bands. Figure 18c shows the parameters of the total active reflection coefficient (TARC) between different ports. The results show that TARC is lower than −20 dB in the working frequency bands. 

As depicted in Figure 19, the minimum and maximum gain in the 3400–3600 MHz frequency band is 1.99 dB and 2.83 dB, respectively. In the frequency band of 5275–5850 MHz, the lowest gain of the antenna is 5.19 dB, and the highest gain reaches 7.4 dB.

Figure 20 shows the efficiency curve of this MIMO antenna, from which the antenna has a minimum efficiency of 58% and a maximum efficiency of 68% in the frequency band of 3400–3600 MHz. In addition, the antenna has a minimum efficiency of 59% and a maximum efficiency of 72% in the band of 5275–5850 MHz. The MIMO antenna has high-efficiency characteristics in the operating frequency bands.

The simulated and measured data of co-pol and cross-pol at different frequency bands are shown in Figure 21 and Figure 22. The co-pol and cross-pol radiation patterns are represented by the dashed and solid lines, respectively. The co-pol radiation of the antenna will dominate the overall radiation effect of the antenna. The cross-pol of the antenna will cause interference with the overall radiation effect of the antenna, which should be minimized as much as possible. From the results, the radiation curve of the co-pol is larger than that of the cross-pol. The actual effect of the antenna is consistent with the simulation results.

The provided MIMO antenna is evaluated against the state-of-the-art broadband 5G MIMO antenna and dual-band MIMO antenna, as outlined in Table 1. This study introduces an innovative broadband 5G MIMO antenna that demonstrates outstanding performance.

## 5. Conclusions

In this paper, an eight-port MIMO antenna suitable for 5G mobile phones is proposed and studied, which covers the n78 (3400–3600 MHz) and WLAN band (5275–5850 MHz). The realization of the dual-frequency band of the antenna is achieved by the folded outer patch and the inner loop. The antenna has a reflection coefficient better than −6 dB and isolation better than 15 dB in the required working frequency band. The antenna also exhibits ideal MIMO performance, and the overall ECC satisfies low MIMO correlation. In addition, the simulation of the effect of the hand on the antenna performance is also carried out, which shows that the MIMO antenna can maintain essential performance in daily use.

## Figures and Tables

**Figure 1 micromachines-14-02200-f001:**
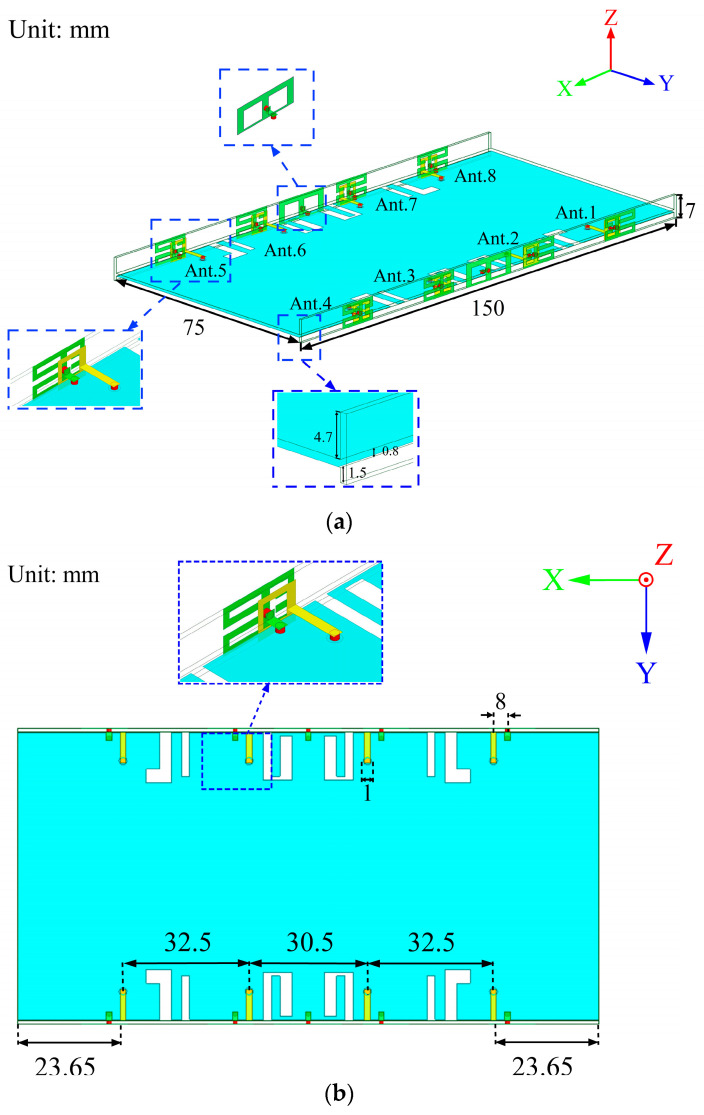
Geometry of the proposed eight-element dual-band MIMO antenna: (**a**) Prospective view; (**b**) top view.

**Figure 2 micromachines-14-02200-f002:**
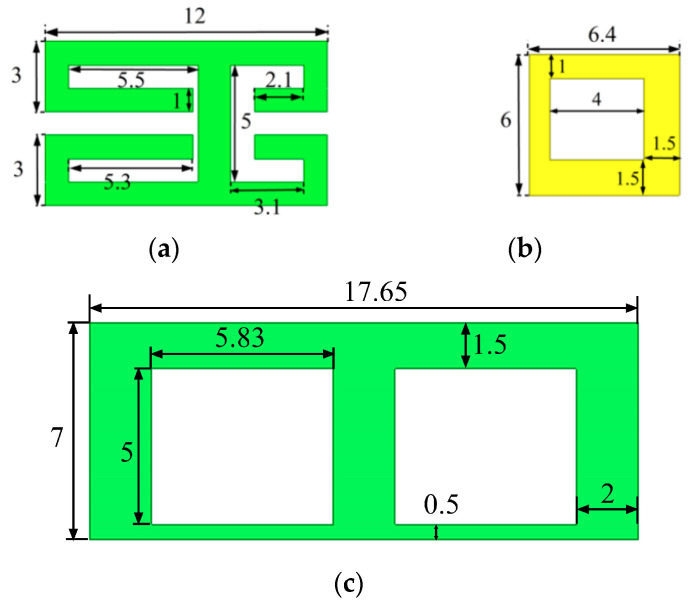
Geometry of the proposed antenna element: (**a**) parasitic unit; (**b**) radiator; (**c**) isolation structure.

**Figure 3 micromachines-14-02200-f003:**
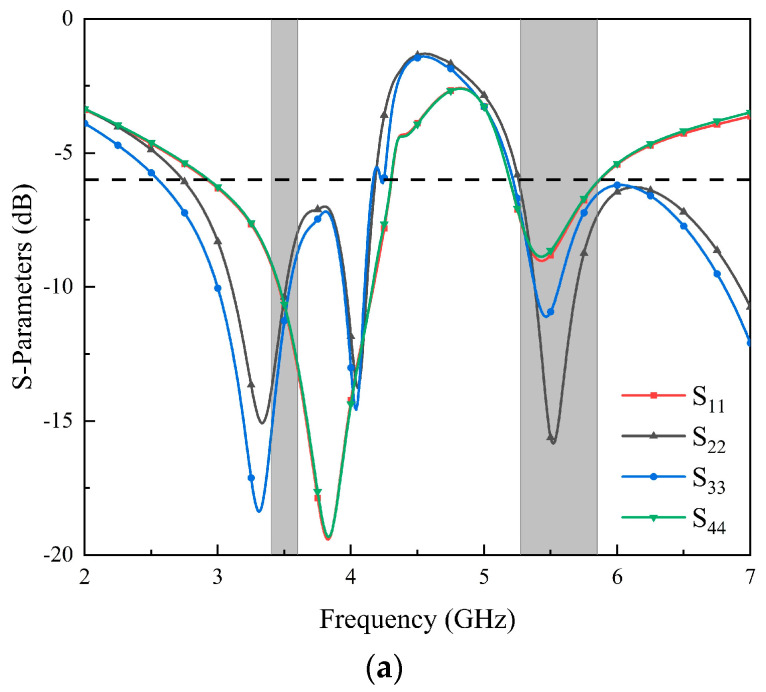
Simulated coefficients of the proposed MIMO antenna: (**a**) the reflection coefficient of Ant. 1 to Ant. 4; (**b**) the transmission coefficient of Ant. 1 to Ant. 5.

**Figure 4 micromachines-14-02200-f004:**
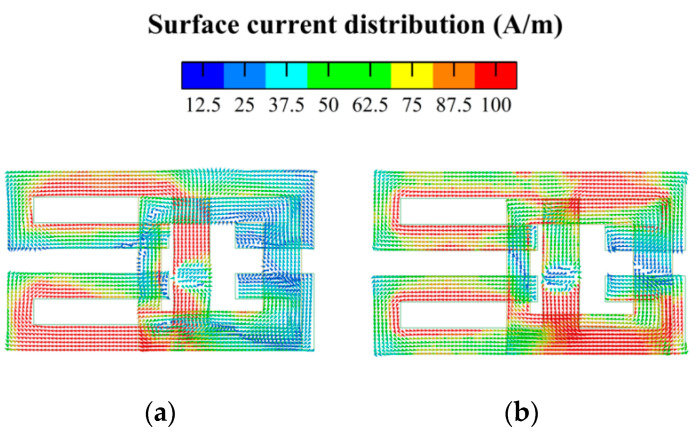
Simulated surface current distribution of antenna element: (**a**) 3.5 GHz; (**b**) 5.5 GHz.

**Figure 5 micromachines-14-02200-f005:**
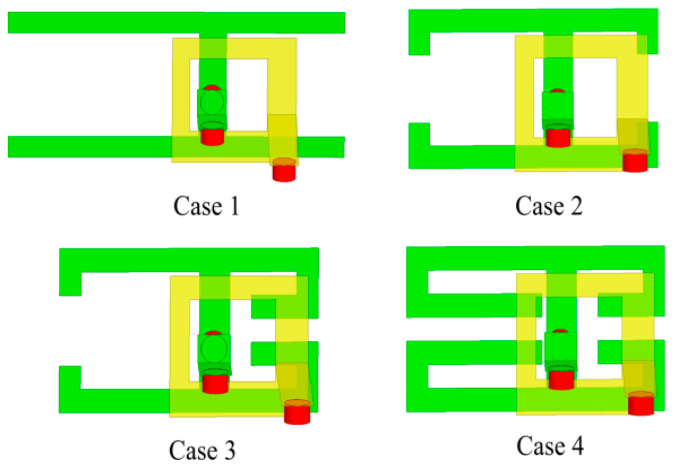
The design evolution process of the antenna element.

**Figure 6 micromachines-14-02200-f006:**
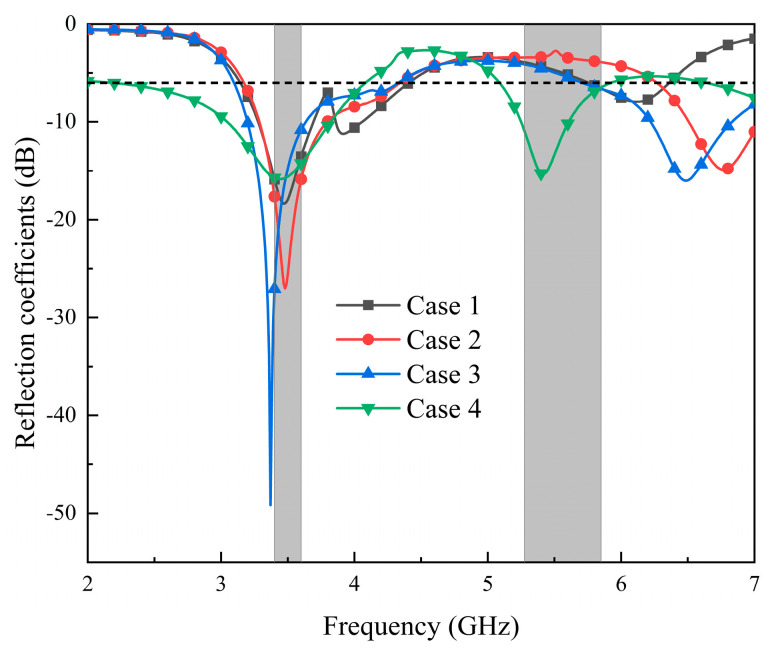
Reflection coefficients of different design processes.

**Figure 7 micromachines-14-02200-f007:**
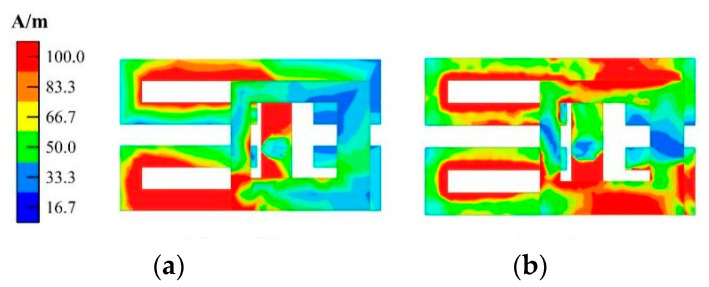
Simulated surface current distribution of the antenna element at: (**a**) 3.5 GHz; (**b**) 5.5 GHz.

**Figure 8 micromachines-14-02200-f008:**
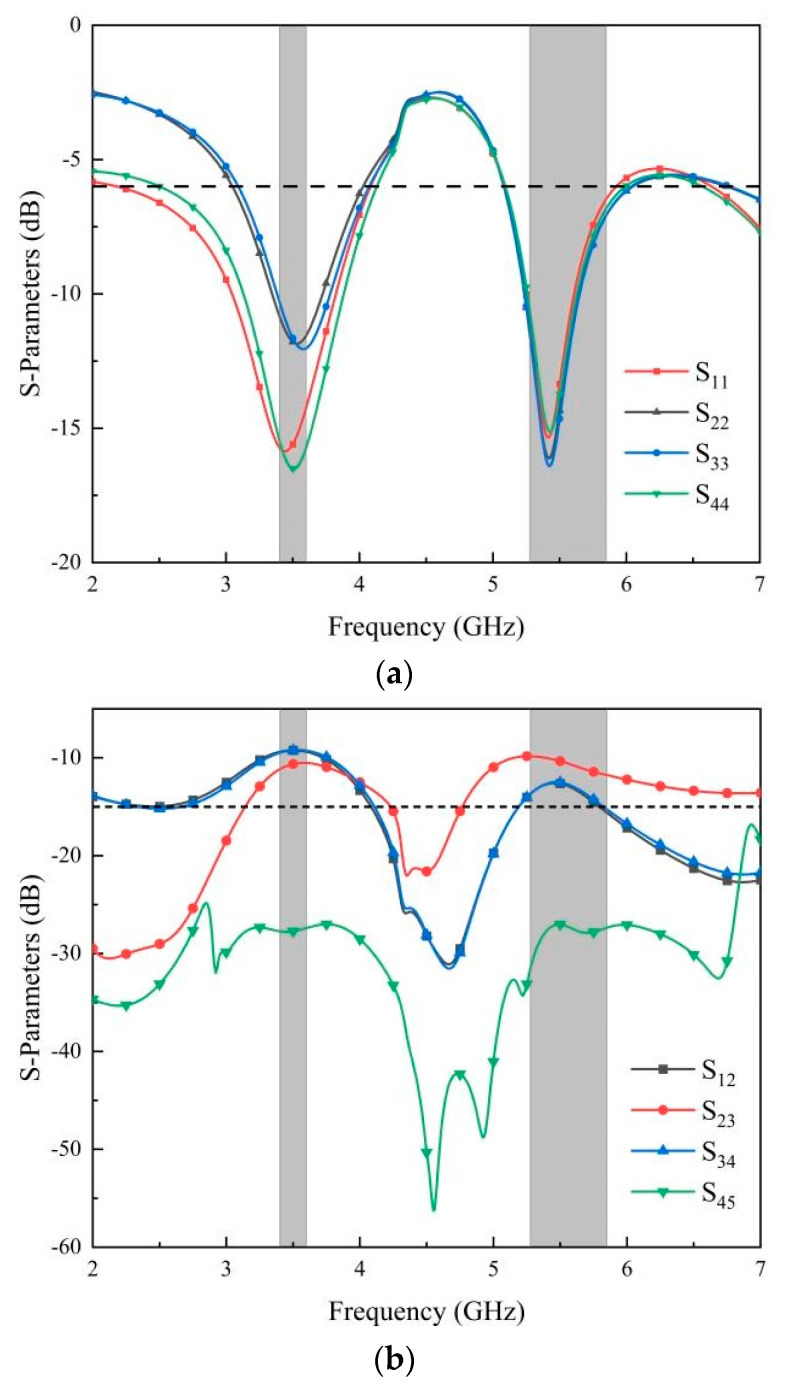
Simulated S-parameters without the isolation structure: (**a**) return loss; (**b**) isolation.

**Figure 9 micromachines-14-02200-f009:**
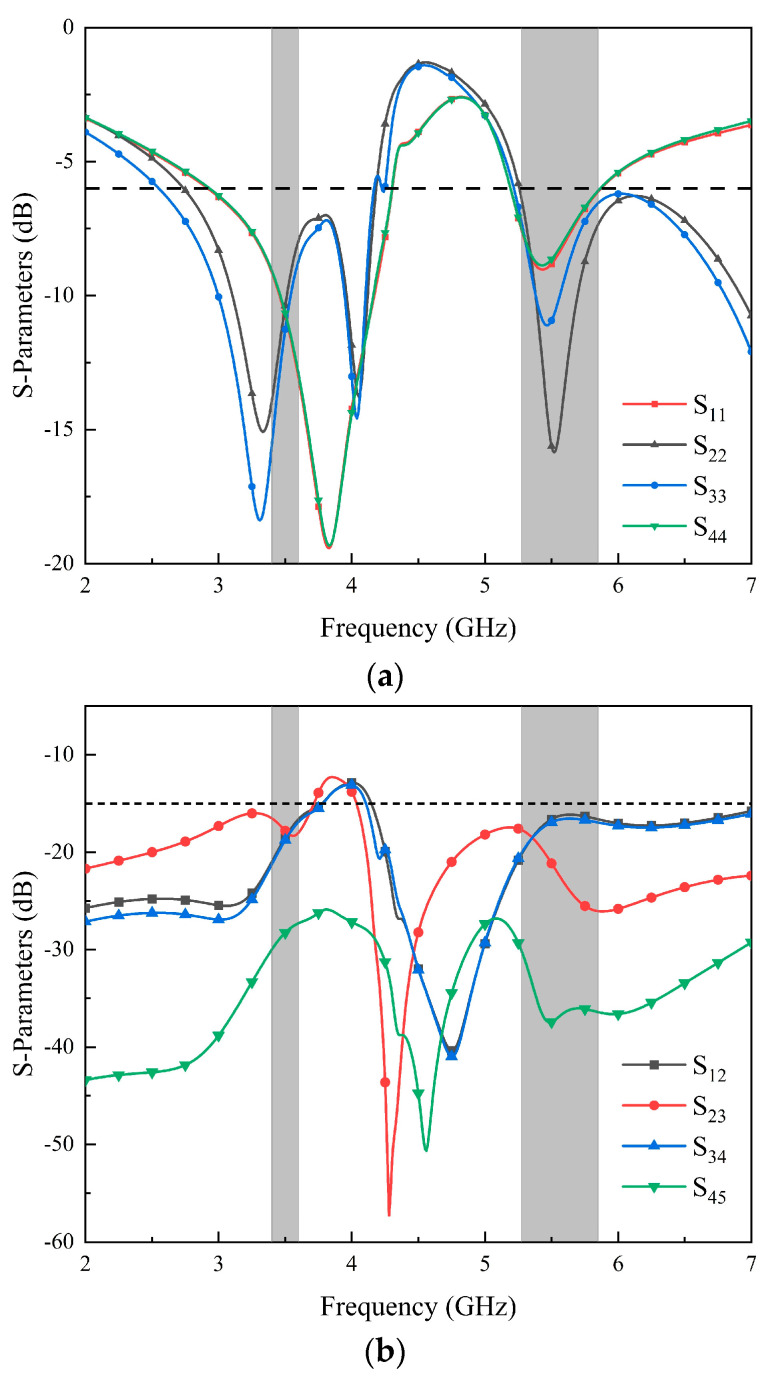
Simulated S-parameters with the isolation structure: (**a**) return loss; (**b**) isolation.

**Figure 10 micromachines-14-02200-f010:**
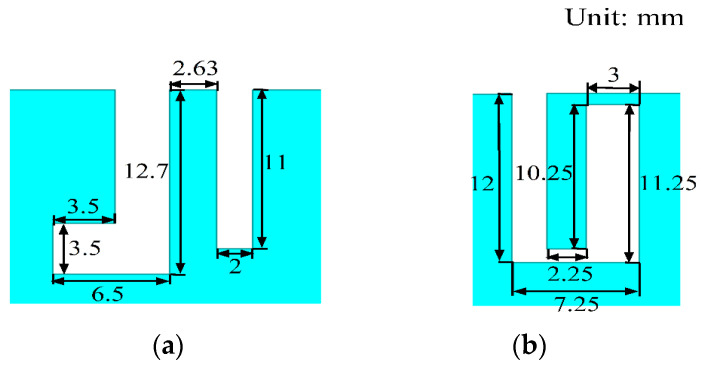
The slots on the top surface of the system board: (**a**) L-shaped slot and rectangular slot; (**b**) U-shaped slot.

**Figure 11 micromachines-14-02200-f011:**
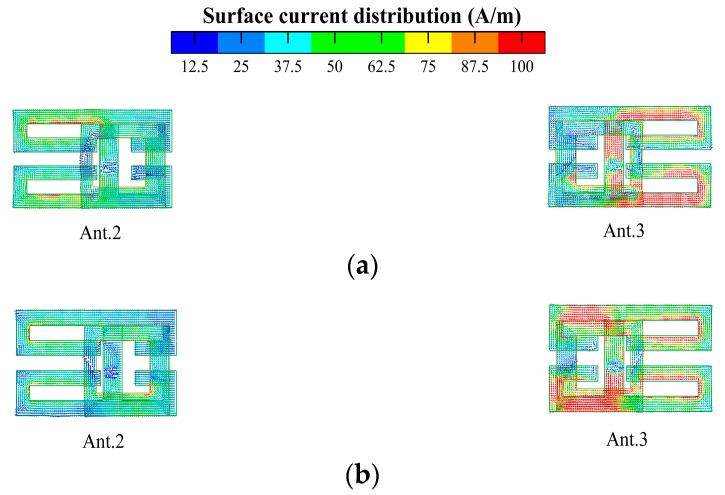
Current distribution without isolation structure at different working frequencies: (**a**) 3.5 GHz; (**b**) 5.5 GHz.

**Figure 12 micromachines-14-02200-f012:**
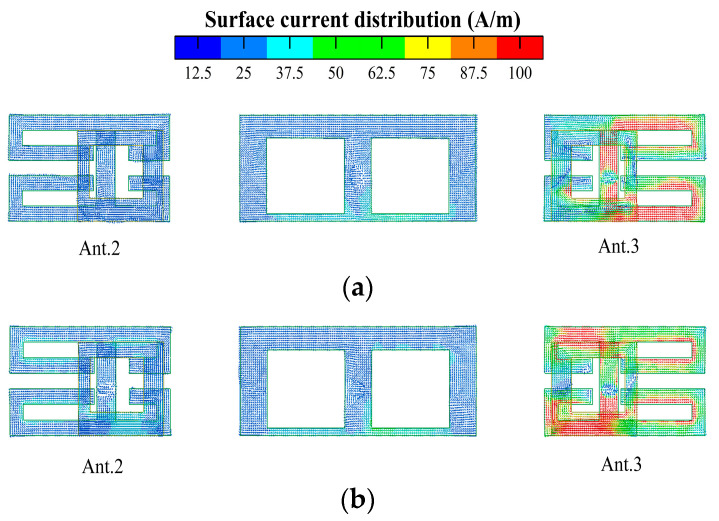
Current distribution with isolation structure (only Ant. 3 is active) at different working frequencies: (**a**) 3.5 GHz; (**b**) 5.5 GHz.

**Figure 13 micromachines-14-02200-f013:**
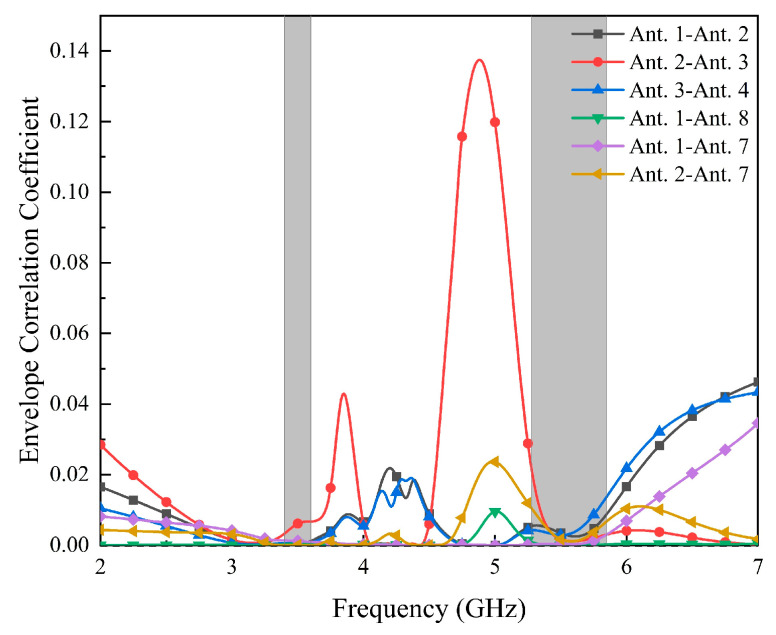
Calculated ECC values.

**Figure 14 micromachines-14-02200-f014:**
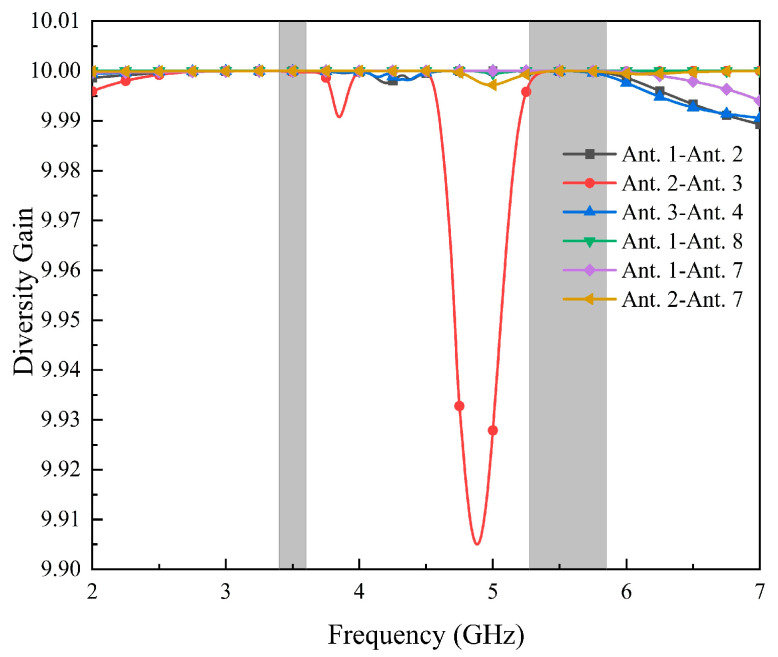
Calculated DG values.

**Figure 15 micromachines-14-02200-f015:**
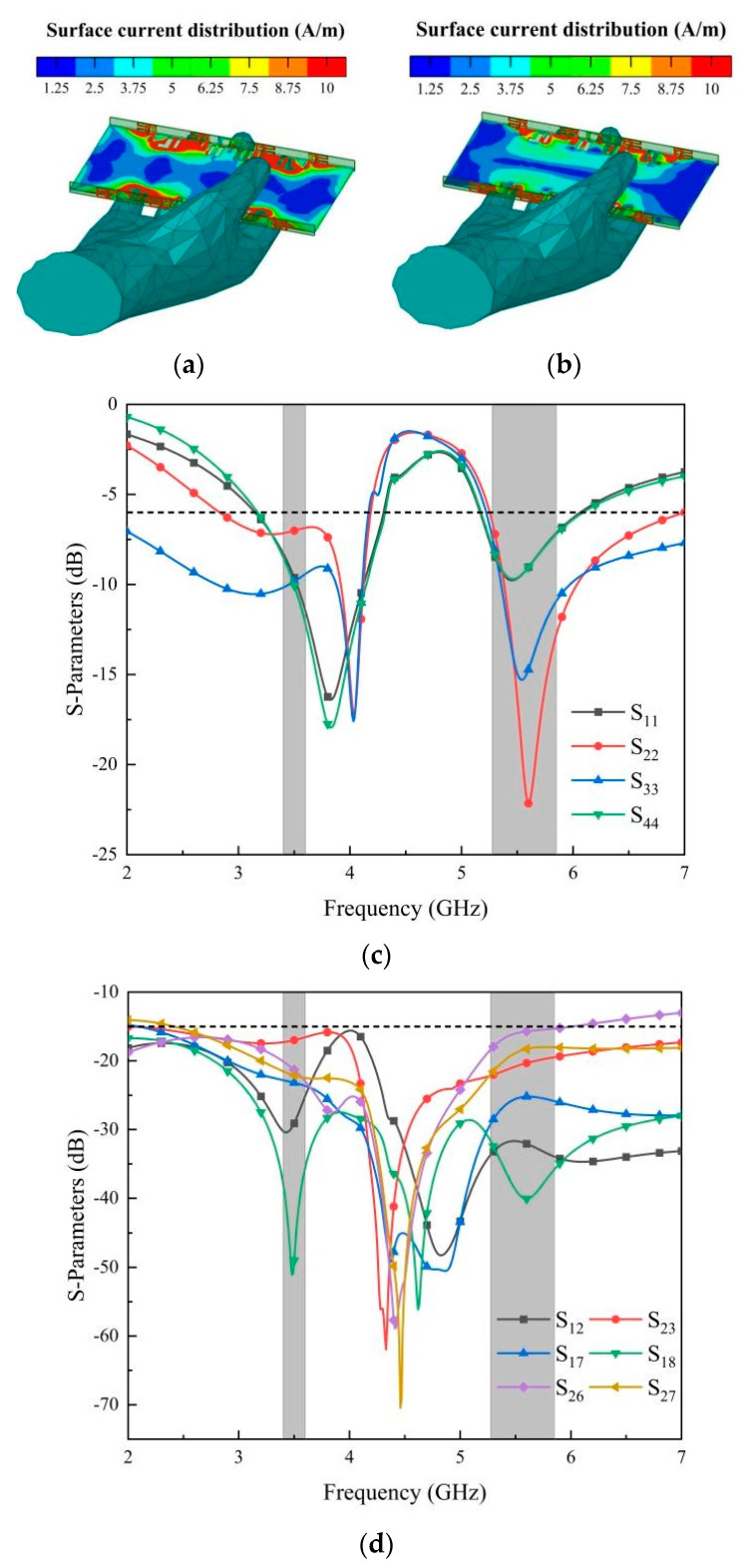
Influence of antenna S-parameters in practical application: (**a**) simulated surface distribution at 3.5 GHz; (**b**) simulated surface distribution at 5.5 GHz; (**c**) simulated results of return loss under the influence of the human body; (**d**) simulated result of isolation under the influence of the human body.

**Figure 16 micromachines-14-02200-f016:**
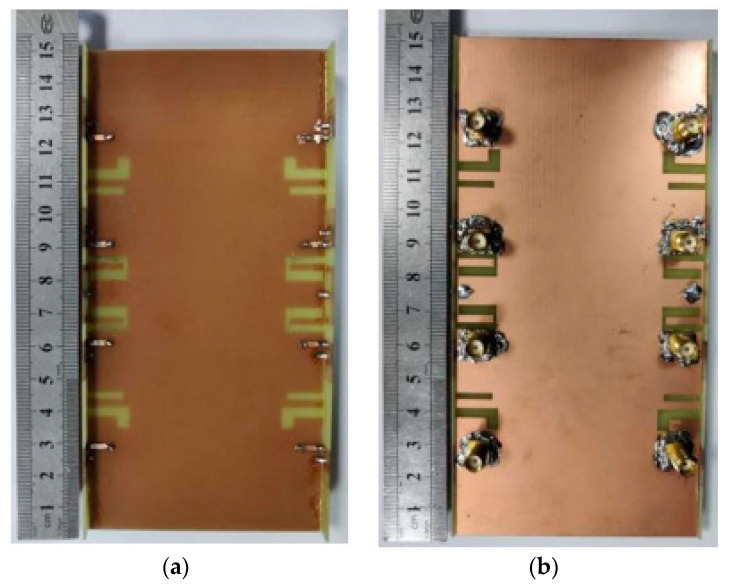
Photographs of the fabricated eight-antenna array: (**a**) top surface; (**b**) bottom surface.

**Figure 17 micromachines-14-02200-f017:**
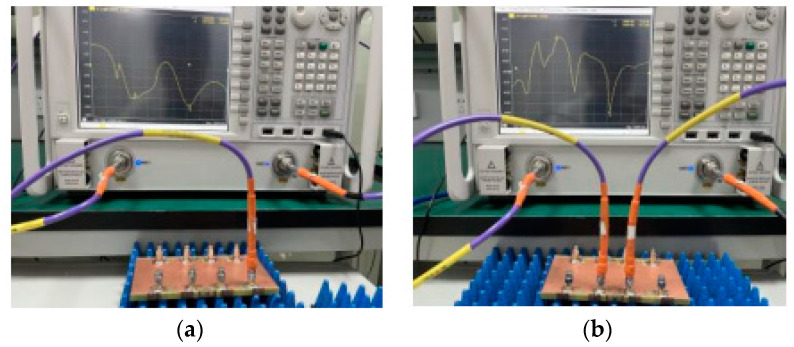
Different antenna port testing: (**a**) test the single port reflection coefficient; (**b**) test the isolation between two ports.

**Figure 18 micromachines-14-02200-f018:**
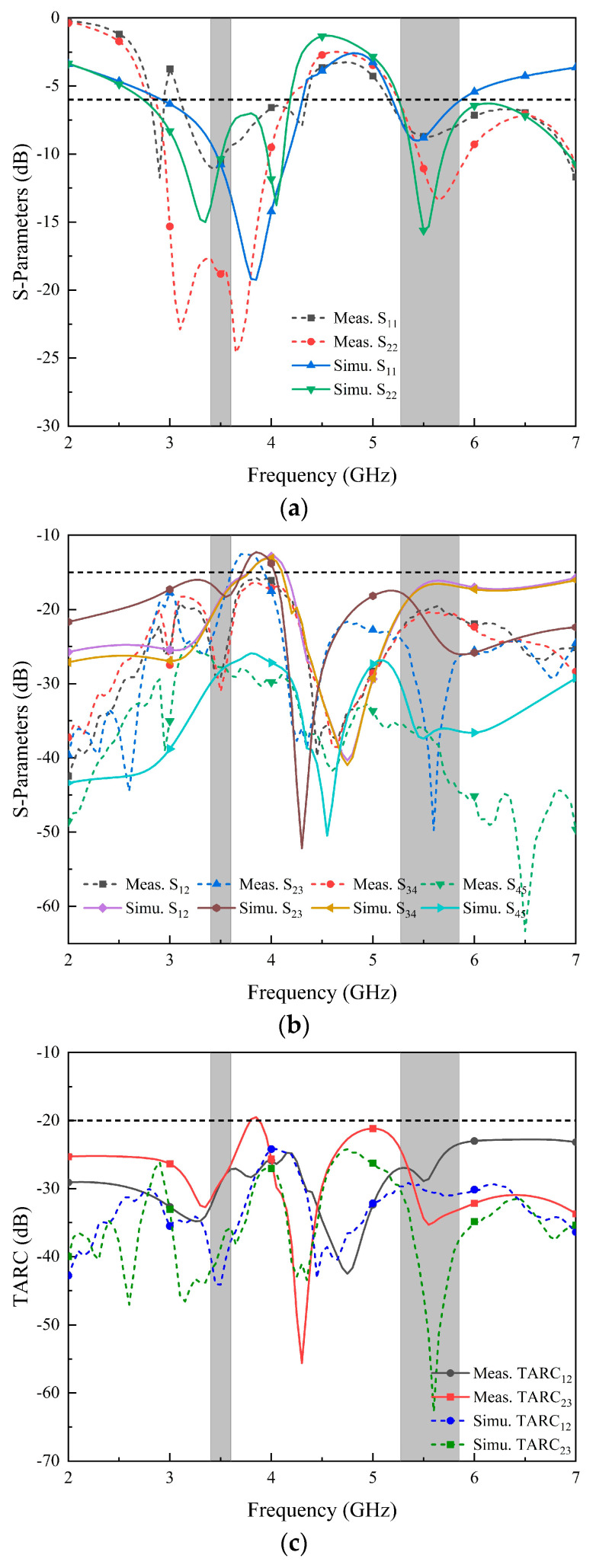
Simulated and measured S-parameters: (**a**) return loss; (**b**) isolation; (**c**) TARC.

**Figure 19 micromachines-14-02200-f019:**
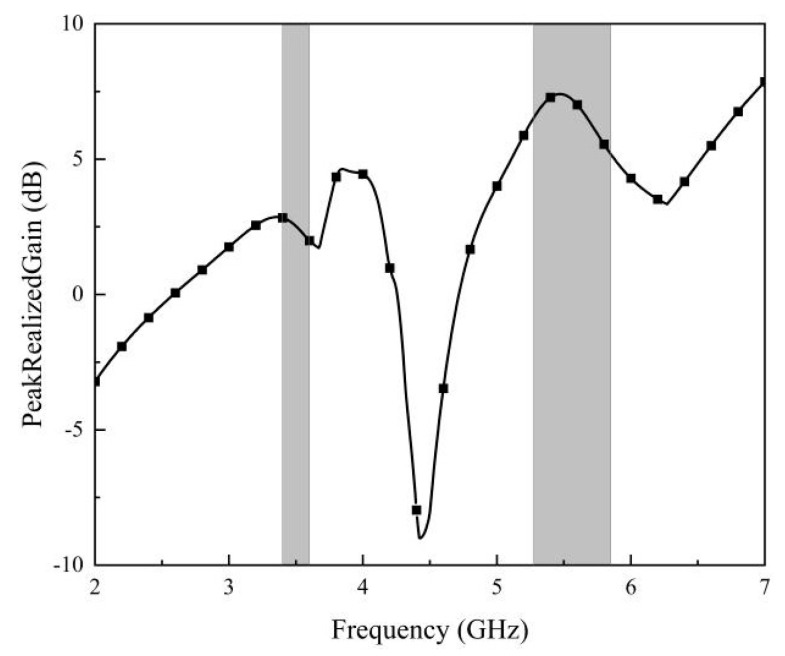
Measured gain of the proposed antenna array.

**Figure 20 micromachines-14-02200-f020:**
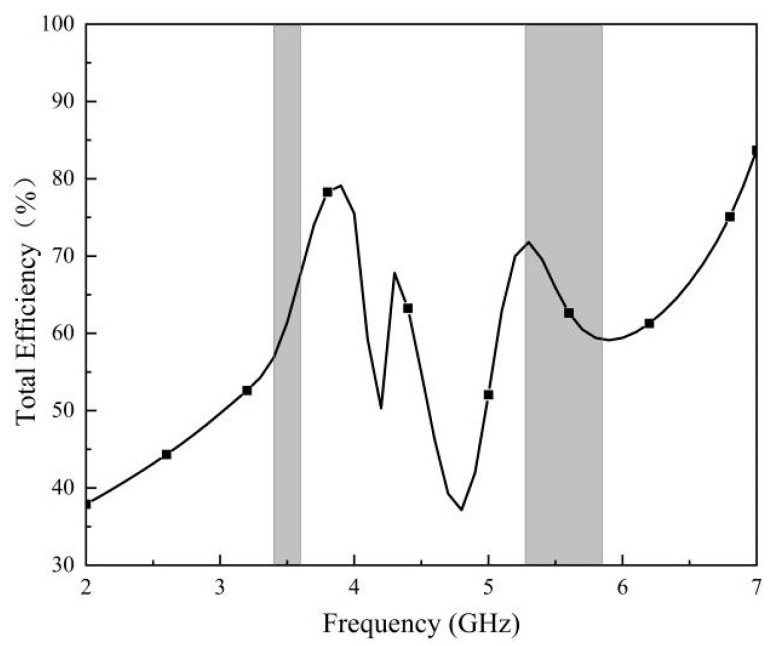
Measured efficiency of the proposed antenna array.

**Figure 21 micromachines-14-02200-f021:**
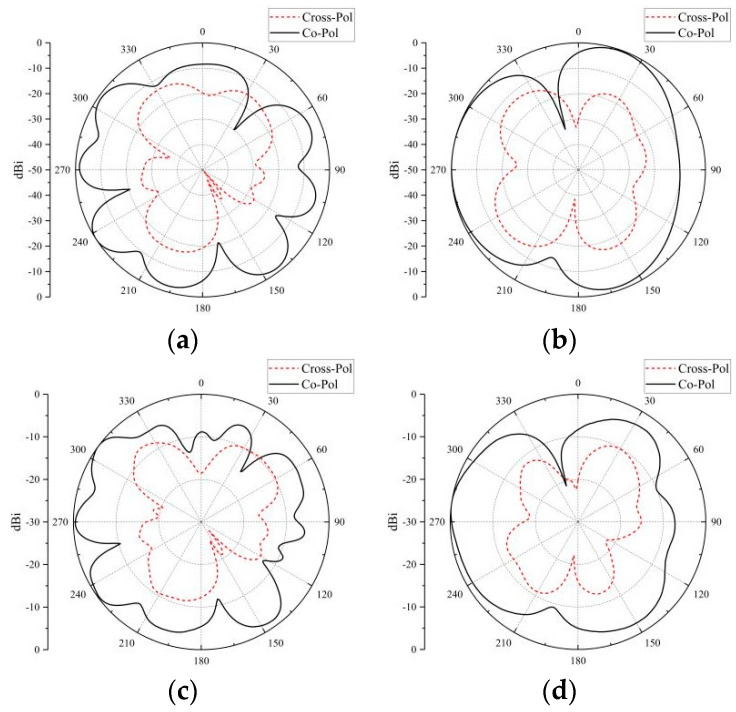
Radiation patterns at 3.5 GHz: (**a**) XOY for simulation; (**b**) YOZ for simulation; (**c**) XOY for measurement; (**d**) YOZ for measurement.

**Figure 22 micromachines-14-02200-f022:**
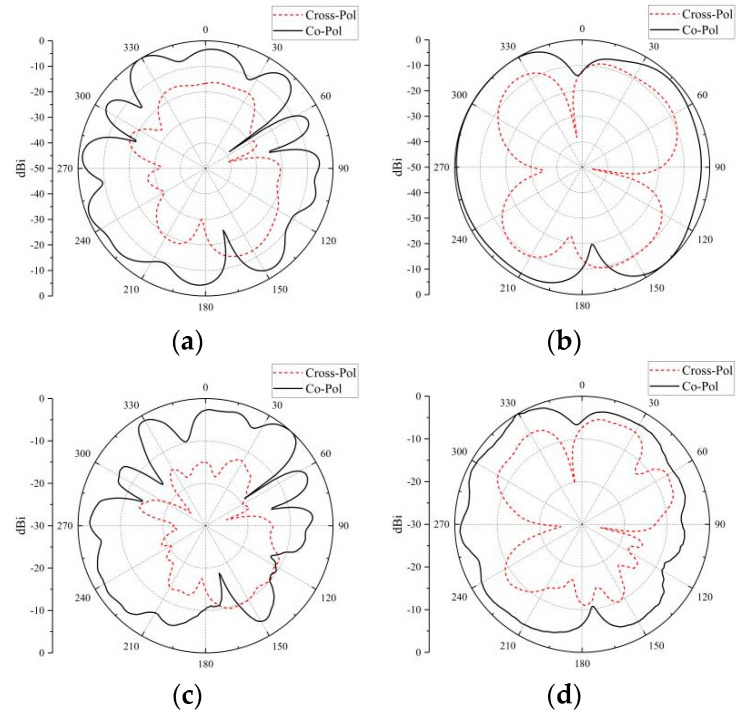
Radiation patterns of 5.5 GHz: (**a**) XOY for simulation; (**b**) YOZ for simulation; (**c**) XOY for measurement; (**d**) YOZ for measurement.

**Table 1 micromachines-14-02200-t001:** Comparison between the proposed and referenced antennas.

Reference	Bandwidth (GHz)	Isolation (dB)	ECC	MIMO Order	Efficiency (%)	Decoupling Method	Size (mm^3^)
[7]	3.4–3.6 (−10 dB)	>15 dB	<0.036	8	62–76	Intrinsic decoupling	150 × 80 × 0.8
[19]	3.4–3.6, 4.8–5.0 (−6 dB)	>12 dB	<0.15	8	>41	Short circuited stub	150 × 64 × 0.8
[20]	3.4–3.6, 5.725–5.925 (−6 dB)	>10 dB	<0.15	8	55–70	Decoupled block	150 × 74 × 7
[21]	3.4–3.6, 4.8–5.1 (−6 dB)	>11.5 dB	<0.08	8	40–85	Neutralization line	150 × 75 × 7
[22]	3.4–3.6, 5.725–5.758 (−6 dB)	>10 dB	<0.2	12	41–59	Self-isolation	150 × 75 × 7
[23]	3.4–3.6, 4.8–5.0 (−10 dB)	>12 dB	<0.06	8	44–72	Self-isolation	150 × 75 × 7
[24]	2.5–3.6 (−10 dB)	>10 dB	<0.2	8	45–64	Spatial decoupling	150 × 75 × 1.6
[25]	3.1–3.9, 5.5–6.3 (−6 dB)	>12 dB	<0.035	8	70–80	Self-isolation	150 × 75 × 7
[26]	3.4–3.6, 5.1–5.7 (−10 dB)	>14 dB	<0.02	8	58–74	Spatial decoupling	150 × 75 × 8
[27]	3.34–3.7, 4.67–5.08 (−6 dB)	>12 dB	<0.08	8	55–72	Slotted decoupling	150 × 75 × 0.8
Proposed	3.4–3.6, 5.275–5.850 (−6 dB)	>15 dB	<0.024	8	58–72	Parasitic strip	150 × 75 × 7

## Data Availability

The data presented in this study are available on request from the corresponding author.

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
