# Peer review of "Eight-Element Dual-Band Multiple-Input Multiple-Output Mobile Phone Antenna for 5G and Wireless Local Area Network Applications"

_micromachines, 2023, doi:10.3390/mi14122200_

Round 1

Reviewer 1 Report

Comments and Suggestions for Authors

This manuscript presents an eight-element dual-band MIMO antenna. Each antenna element consists of a radiator on the internal surface and a defected ground plane on the outer surface can cover the N78 and WiFi-5G bands. Open slot and parasitic element are applied to reduce the mutual coupling. The results are acceptable.

1. The antenna size, efficiency, decoupling method should be added in the comparisons table to show the merits.

2. The antenna structure is not clear. The authors must improve the figure.

3. The decoupling methods of the slot and parasitic element have been widely studied in the open literature. Please cite this method in the manuscript.

[Ref A] A Wideband Quad-Antenna System for Mobile Terminals, IEEE AWPL, 2014.

[Ref B] A Compact Uniplanar Printed Dual-Antenna Operating at the 2.4/5.2/5.8 GHz WLAN Bands for Laptop Computers, IEEE AWPL, 2014.

4. The effect of the defected ground plane and isolation structure on the S21 should be analyzed separately. Also, the isolation should the positive value.

Author Response

Point 1: The antenna size, efficiency, decoupling method should be added in the comparisons table to show the merits.

Response 1: We have added all the required contents to the comparison table that is Table 1.

Point 2: The antenna structure is not clear. The authors must improve the figure.

Response 2: We have modified Fig.1 (a) and Fig.1 (b) .

Point 3: The decoupling methods of the slot and parasitic element have been widely studied in the open literature. Please cite this method in the manuscript.

Response 3: We have added the following two references and descripted the decoupling methods from line 66 to 69. [17] Wang, Y.; Du, Z. A Wideband Quad-Antenna System for Mobile Terminals. IEEE Antennas and Wireless Propagation Letters. 2014, 13, 1521-1524. [18] Guo, L.; Wang, Y.; Du, Z.; Gao, Y.; Shi, D. A Compact Uniplanar Printed Dual-Antenna Operating at the 2.4/5.2/5.8 GHz WLAN Bands for Laptop Computers. IEEE Antennas and Wireless Propagation Letters. 2014, 13, 229-232.

Point 4: The effect of the defected ground plane and isolation structure on the S21 should be analyzed separately. Also, the isolation should the positive value.

Response 4: The isolation of the antenna is discussed in detail from line 204 to 206. The function and characteristics of each structure are described.

Reviewer 2 Report

Comments and Suggestions for Authors

This paper presents an eight-element MIMO antenna for 5G and WLAN applications. This paper is informative and well-organized. But I have one small comment. 

In Fig. 26, when the authors conducted different port antenna tests, the other ports that were not activated by VNA needed to be terminated by perfectly matched loads. 

Author Response

Point 1: In Fig. 26, when the authors conducted different port antenna tests, the other ports that were not activated by VNA needed to be terminated by perfectly matched loads.

Response 1: As shown in Fig.17, all the other ports are terminated by a 50Ω load. There are only a few variation of the measurement results which we have updated.

Reviewer 3 Report

Comments and Suggestions for Authors

1. Authors proposed Eight Element Dual Band MIMO antenna for 5G and WLAN Applications. The proposed structure is targeting mobile phones. However, the title doesn't give this impression. It is recommended to include "mobile phone" or other relevant term in the title.

2. Authors said that they used internal and outer frame of the antenna to reduce the internal space of mobile phone. Radiator is on internal surface and defects on ground is on outer surface. How about isolation structure as mentioned in Fig. 2c? could you please justify this part of the antenna?

3.  It is visually confusing to understand the evolution of antenna while looking at Fig. 5. It is better to improve this figure. Also, what are these red dots? what it represents in Fig. 5? 

4. It is recommended to add ECC and efficiency columns in Table 1 and compare with other reported work.

Comments on the Quality of English Language

The manuscript demonstrates good written English quality.

Author Response

Point 1: Authors proposed Eight Element Dual Band MIMO antenna for 5G and WLAN Applications. The proposed structure is targeting mobile phones. However, the title doesn't give this impression. It is recommended to include "mobile phone" or other relevant term in the title.

Response 1: We have revised the title.

Point 2: Authors said that they used internal and outer frame of the antenna to reduce the internal space of mobile phone. Radiator is on internal surface and defects on ground is on outer surface. How about isolation structure as mentioned in Fig. 2c? could you please justify this part of the antenna?

Response 2: We have investigated the function of the isolation structure in line 206 to 208. The current distributions of the isolation structure are shown in Fig. 12.

Point 3: It is visually confusing to understand the evolution of antenna while looking at Fig. 5. It is better to improve this figure. Also, what are these red dots? what it represents in Fig. 5?

Response 3: The vias connected to the parasitic units and the feed lines are represented by red cylinders in Figure 5, as demonstrated on lines 140 to 141.

Point 4: It is recommended to add ECC and efficiency columns in Table 1 and compare with other reported work.

Response 4: According to your requirements, the corresponding content has been added to the comparison table.

Round 2

Reviewer 3 Report

Comments and Suggestions for Authors

Authors have improved the revised manuscript according to the said comments.